# Severe Acute Kidney Injury in Hospitalized Cancer Patients: Epidemiology and Predictive Model of Renal Replacement Therapy and In-Hospital Mortality

**DOI:** 10.3390/cancers16030561

**Published:** 2024-01-28

**Authors:** Roberto Calças Marques, Marina Reis, Gonçalo Pimenta, Inês Sala, Teresa Chuva, Inês Coelho, Hugo Ferreira, Ana Paiva, José Maximino Costa

**Affiliations:** 1Nephrology Department, Centro Hospitalar Universitário do Algarve, 8000 Faro, Portugal; 2Nephrology Department, Centro Hospitalar Universitário de Coimbra, 3004 Coimbra, Portugal; 3Nephrology Department, Centro Hospitalar de Lisboa Ocidental, 2790 Lisboa, Portugal; 4Nephrology Department, Centro Hospitalar Universitário de Santo António, 4050 Porto, Portugal; 5Nephrology Department, Instituto Português de Oncologia do Porto, 4200 Porto, Portugal

**Keywords:** acute kidney injury, cancer, epidemiology, mortality, renal replacement therapy

## Abstract

**Simple Summary:**

Acute kidney injury (AKI) is a common complication among hospitalized cancer patients, impacting the effectiveness of anticancer treatment and being associated with a poor prognosis. The identification of risk factors and the underlying cause(s) of AKI ensures appropriate intervention. Limited studies on this subject have been conducted to identify high-risk patients and guide decisions on the initiation of renal replacement therapy (RRT). Our study provides an overview of the causes of AKI and identifies prognostic determinants of the need for RRT and in-hospital mortality. It introduces an easily calculated risk score that combines acute risk factors to predict in-hospital mortality, potentially helping in clinical practice with the complex decision to initiate or forgo RRT.

**Abstract:**

Background: Acute kidney injury (AKI) is a common complication among cancer patients, often leading to longer hospital stays, discontinuation of cancer treatment, and a poor prognosis. This study aims to provide insight into the incidence of severe AKI in this population and identify the risk factors associated with renal replacement therapy (RRT) and in-hospital mortality. Methods: This retrospective cohort study included 3201 patients with cancer and severe AKI admitted to a Comprehensive Cancer Center between January 1995 and July 2023. Severe AKI was defined according to the KDIGO guidelines as grade ≥ 2 AKI with nephrological in-hospital follow-up. Data were analyzed in two timelines: Period A (1995–2010) and Period B (2011–2023). Results: A total of 3201 patients (1% of all hospitalized cases) were included, with a mean age of 62.5 ± 17.2 years. Solid tumors represented 75% of all neoplasms, showing an increasing tendency, while hematological cancer decreased. Obstructive AKI declined, whereas the incidence of sepsis-associated, prerenal, and drug-induced AKI increased. Overall, 20% of patients required RRT, and 26.4% died during hospitalization. A predictive model for RRT (AUC 0.833 [95% CI 0.817–0.848]) identified sepsis and hematological cancer as risk factors and prerenal and obstructive AKI as protective factors. A similar model for overall in-hospital mortality (AUC 0.731 [95% CI 0.71–0.752]) revealed invasive mechanical ventilation (IMV), sepsis, and RRT as risk factors and obstructive AKI as a protective factor. The model for hemato-oncological patients’ mortality (AUC 0.832 [95% CI 0.803–0.861]) included IMV, sepsis, hematopoietic stem cell transplantation, and drug-induced AKI. Mortality risk point score models were derived from these analyses. Conclusions: This study addresses the demographic and clinical features of cancer patients with severe AKI. The development of predictive models for RRT and in-hospital mortality, along with risk point scores, may play a role in the management of this population.

## 1. Introduction

Recent advances in cancer diagnosis and treatment have substantially improved outcomes for oncological patients [1,2]. Concurrently, there is an increasing recognition of cancer-related complications, including acute kidney injury (AKI), with an overall incidence varying from 7.5 to 25.8% in large series [3,4,5].

The occurrence of AKI in this population may be directly related to malignancy, resulting from the direct effects of cancer therapy that arise from treatment complications, or be associated with non-cancer-related risk factors [6]. This multifactorial condition confers a poor prognosis, often leading to an extended hospital stay and compromising the efficacy of anticancer treatments [7]. The latter may occur because of the need to discontinue or adjust targeted drug doses. Consequently, there is a risk of overdosing, leading to increased toxicity, or suboptimal dosing, reducing the likelihood of achieving remission [8,9].

An important and potentially effective strategy in AKI management is to uncover its underlying cause(s) and associated risk factors to ensure appropriate intervention [10,11]. A subset of patients, typically characterized by more severe clinical presentations and prognoses, may require nephrological consultation. The integration of prognostic tools that identify individuals at high risk of mortality may help in the decision to initiate or forgo renal replacement therapy (RRT) [12].

Previous studies on AKI in cancer patients have been limited to specific patient groups, and high-quality epidemiological data are lacking to help with thoughtful decision-making. Our study aimed to address this gap by determining the incidence of severe AKI, providing an overview of the most common causes, and identifying prognostic determinants of the need for RRT and in-hospital mortality in a Comprehensive Cancer Center. Our objective was also to develop a quantitative risk score for in-hospital mortality in patients with severe AKI.

## 2. Materials and Methods

### 2.1. Study Population and Data Collection

This retrospective cohort study of prospectively collected data, conducted at a single tertiary referral oncological center, included patients with severe AKI (stage 2 or higher according to the Kidney Disease: Improving Global Outcomes (KDIGO) criteria) who were admitted to our inpatient clinic between 1 January 1995, and 31 July 2023, and with nephrological in-hospital follow-up. Data analysis involved a comparison between two timelines: 1995–2010 (referred to as Period A) and 2011–2023 (referred to as Period B), corresponding to the era of conventional chemotherapy and the exponential utilization of molecularly targeted agents, respectively.

Patients without an indication for cardiopulmonary resuscitation and those for whom palliative care was the only cancer treatment option were excluded. Patients who underwent renal transplantation were also excluded. 

Laboratory and clinical information were extracted from both paper and electronic hospitalization databases.

Data collection included basic demographic characteristics (age, sex), cancer diagnosis, etiological characterization of AKI, postoperative state, intensive care unit (ICU) admission, and need for invasive mechanical ventilation (IMV). Regarding hematological malignancies, information on treatment with hematopoietic stem cell transplantation (HSCT), specifying the type of donor and graft-versus-host disease (GvHD), was also gathered. The outcomes of interest included the need for RRT and in-hospital mortality.

Approval was obtained from the Instituto Português de Oncologia do Porto Ethics Committee.

### 2.2. Identification and Classification of Severe AKI

Severe AKI was defined as an increase in serum creatinine (SCr) of more than twice the baseline SCr within 7 days, as outlined by the KDIGO guidelines [13]. Urine output was not considered because it was unavailable to a substantial proportion of the patients.

In most cases, the baseline SCr was considered the lowest SCr level obtained during the hospitalization period or from previous assessments. For individuals with previously normal renal function, we relied on creatinine measurements obtained within the year prior to hospital admission. For patients with chronic kidney disease (CKD), this parameter was limited to determinations obtained within the preceding 3 months.

The AKI stage was determined using the peak SCr level after AKI diagnosis and classified according to the KDIGO criteria.

### 2.3. AKI Etiology

For each case, the nephrologist was asked to identify the potential primary underlying cause of AKI based on the following classifications: exclusively prerenal, obstructive, sepsis-associated, drug-induced, hypercalcemia, tumor lysis syndrome (TLS), glomerulopathy, thrombotic microangiopathy, leukemic infiltration, myeloma cast nephropathy, monoclonal gammopathy of renal significance (MGRS), and GvHD.

Prerenal AKI was characterized by evidence of volume depletion related to gastrointestinal symptoms (nausea, vomiting, and diarrhea), reduced food intake, bleeding, and/or anemia [10]. It also encompasses nephrectomy-induced ischemic injury and edematous states (hepatorenal and cardiorenal syndromes). A presumed subgroup of patients within this classification may have developed ischemic acute tubular necrosis due to prolonged prerenal injury.

Sepsis-associated AKI was defined as AKI in patients with sepsis or septic shock according to the Third International Consensus Definitions for Sepsis and Septic Shock (Sepsis-3) [14].

Owing to the multifactorial nature of AKI, more than one main cause was attributed in some cases.

### 2.4. Statistical Analysis

Categorical variables are presented as frequencies and percentages, whereas continuous parametric variables are expressed as the mean and standard deviation (SD). Univariate analysis was performed using the chi-square test to examine the association between each independent variable and both the need for RRT and in-hospital mortality.

Logistic regression was used to estimate the odds ratios (ORs) of the risk factors influencing the requirement for RRT and in-hospital mortality within the cohort. All variables showing significance in univariate analysis (*p* < 0.20) were incorporated into a backward stepwise logistic regression model. Terms were eliminated if the likelihood ratio statistics had a *p* > 0.10 and reintroduced if the likelihood ratio statistic indicated a *p* < 0.05.

An additive risk score for the predicted probability of in-hospital mortality was created by attributing points corresponding to the nearest integer of twice each covariate’s OR.

Discrimination and calibration were assessed to evaluate the logistic regression and risk score models. Discrimination was evaluated by the area under the receiver operating characteristic (ROC) curve, with a value > 0.7 considered indicative of sufficient predictive accuracy. The optimal cut-off value between sensitivity and specificity was determined using Youden’s index (sensitivity + specificity − 1). Calibration, the agreement between the predicted and observed risk of death, was assessed using the Hosmer–Lemeshow goodness-of-fit test.

Statistical analysis was performed using the ‘Statistical Package for the Social Sciences’ version 29.0 for Windows (SPSS, Chicago, IL, USA). Statistical significance was set at *p* < 0.05.

## 3. Results

### 3.1. Baseline Characteristics

A total of 3201 cancer patients with severe AKI were included in the study. The mean age was 62.5 ± 17.2 years, and 57.4% (*n* = 1836) were male. Baseline characteristics of the patients are presented in Table 1.

Solid tumors comprised 75.7% of all neoplasms, with gastrointestinal, urological, and gynecological tumors being the most prevalent (30.9% (n = 749), 23.6% (n = 571), and 18.1% (n = 438)). Less common cancer types included breast cancer (7.1%, n = 172), head and neck cancer (5.3%, n = 128), lung cancer (5.1%, n = 123), sarcoma (4.7%, n = 115), and other solid neoplasms (5.2%, n = 126). There was an increasing trend in solid tumors (80.6% vs. 70.7%, *p* < 0.001).

Non-Hodgkin lymphoma was the most prevalent hematological cancer (29.5%, n = 230), followed by multiple myeloma (23.6%, n = 184), acute myeloid leukemia (17.0%, n = 132), and acute lymphoid leukemia (13.7%, n = 107). Other diagnoses (16.2%, n = 126) included myelodysplastic syndrome, chronic myeloid leukemia, chronic lymphoid leukemia, Hodgkin lymphoma, and aplastic anemia. Of the 280 (36.5%) patients undergoing HSCT, 149 (52.5%) received autologous transplantation and 135 (47.5%) received allogeneic transplantation, with 65.9% (n = 89) of allogeneic cases developing GvHD.

### 3.2. Incidence and Characterization of Severe AKI

Between January 1995 and July 2023, 315,932 patients were hospitalized, corresponding to an overall incidence of severe AKI of 1.0%.

The most common cause of severe AKI was obstructive (24.2%, n = 817), followed by sepsis-associated (22.9%, n = 773), exclusively prerenal (19.7%, n = 666), and drug-induced AKI (17.7%, n = 599). These represented more than 80% of all causes. Further details of the additional causes are presented in Figure 1.

When comparing the four most common etiologies (Table 2), obstructive AKI showed a higher association with urological (32.9%) and gynecological cancers (31.8%), and prerenal AKI was more commonly linked to gastrointestinal tumors (35.3%). In contrast, sepsis- and drug-induced AKI were predominantly associated with hematological neoplasms (43.5% and 40.4%, respectively).

Despite being the most common AKI etiology, the incidence of obstructive AKI has significantly decreased in recent years (19.5% vs. 32%, *p* < 0.001). In contrast, an increasing incidence of other causes was reported, including sepsis-associated AKI (28.8% vs. 19.4%, *p* < 0.001) and prerenal AKI (24.1% vs. 17.6%, *p* < 0.001).

The incidence of postoperative AKI was 10% (n = 321). The four most common neoplasms in this setting were gastrointestinal (45.2%, n = 145), urological (19%, n = 61), gynecological (12.5%, n = 40), and head and neck cancer (10%, n = 32). 

Concerning drug-induced AKI, its incidence has significantly increased in recent years (20.7% vs. 16.8%, *p* = 0.006), concurrent with the emergence of AKI attributable to immunotherapy (3.2% vs. 0%, *p* < 0.001). Approximately one-third of the patients had more than one drug identified as potentially contributing to AKI. Notably, antineoplastic treatment accounted for only 17.8% of all cases, with platinum-derived agents being the most frequently implicated class (n = 54), followed by immune checkpoint inhibitors (ICI; n = 52), ifosfamide (n = 12), and tyrosine kinase inhibitors (TKI; n = 6). The predominant drugs identified were not directly related to oncological treatment, with antibiotics (n = 193) ranking as the most frequent, followed by nonsteroidal anti-inflammatory drugs (n = 151), antivirals (n = 84), and antifungals (n = 80). Less frequently implicated drugs included calcineurin inhibitors (n = 49) and bisphosphonates (n = 15). Contrast media nephrotoxicity was reported in 46 patients, and it was mainly related to intra-arterial tumor embolization.

### 3.3. Need for RRT

Overall, 20% (n = 639) of patients required RRT: continuous RRT (CRRT) in 51% (n = 328), intermittent hemodialysis (IHD) in 43% (n = 274), and both modalities in 6% (n = 36). This number increased over the last years (22.3% vs. 17.6%, *p* = 0.002).

In univariate analysis, male sex (*p* = 0.008), age < 40 years (*p* < 0.001), hematological cancer (*p* < 0.001), ICU admission (*p* < 0.001), and IMV (*p* < 0.001) were associated with higher probability of requiring RRT. Concerning AKI etiology, sepsis and drug-induced AKI were more frequently associated with RRT (*p* < 0.001), whereas exclusively prerenal and obstructive AKI showed a less frequent association (*p* < 0.001). When analyzing exclusively the hemato-oncological patients, both autologous and allogeneic HSCT required RRT more frequently (*p* < 0.001), as well as those with GvHD (*p* < 0.001). 

In the multivariate analysis (Table 3), IMV (OR 7.450 [95% CI 5.455–10.174]) and hematologic cancer (OR 2.325 [95% CI 1.849–2.925]) were identified as predictive factors associated with the need for RRT. Conversely, prerenal (OR 0.019 [95% CI 0.006–0.060]) and obstructive AKI (OR 0.339 [95% CI 0.251–0.457]) were protective factors against the need for RRT. In this multivariate model, sepsis-associated AKI did not reach statistical significance (OR 1.279 [95% CI 0.980–1.670]). Upon excluding IMV from the model (Table 4), sepsis-associated AKI emerged as a robust predictor of RRT requirement (OR 3.037 [95% CI 2.462–3.746]).

The reliability of both models was adequate, with *p*-values of 0.565 and 0.084, respectively. The AUC was 0.854 (95% CI 0.838–0.869) for the first model and 0.833 (95% CI 0.817–0.849) for the second model, indicating good discriminability in both cases.

Table 5 shows the multivariate model specifically for patients with hematological cancer (*p* = 0.174 for the Hosmer–Lemeshow test, AUC of 0.798, 95% CI 0.767–0.830), revealing that IMV (OR 11.533 [95% CI 7.954–16.723]) and age < 40 years (OR 1.501 [95% CI 1.001–2.250]) were predictors for RRT, while prerenal (OR 0.056 [95% CI 0.013–0.247]) and obstructive AKI (OR 0.457 [95% CI 0.245–0.854]) were identified as protective factors.

### 3.4. In-Hospital Mortality and Risk Score Model

During hospitalization, 846 (26.4%) patients died. In-hospital mortality decreased over time (23.7% vs. 29.2%, *p* < 0.001). Demographic and clinical data were compared between survivors and in-hospital deceased patients (Table 6). CRRT patients had an almost five times higher mortality risk than IHD patients (OR 8.545 [95% CI 6.717–10.871] vs. OR 1.756 [95% CI 1.375–2.244].

In the multivariate logistic regression analysis, IMV (OR 3.804 [95% CI 2.844–5.089]), sepsis-associated AKI (OR 2.164 [95% CI 1.731–2.706]), and the need for RRT (OR 2.046 [95% CI 1.640–2.552]) were predictors of in-hospital death, while obstructive AKI was found to be protective (OR 0.647 [95% CI 0.518–0.809]). The attributable scores for each variable are listed in Table 7.

Both models demonstrated adequate reliability (*p* = 0.834 and *p* = 0.904, respectively) with overlapping AUC, indicating acceptable discriminability (Figure 2). According to the best Youden’s index, the optimum cut-off value of 4 predicted in-hospital mortality using this risk score model with a sensitivity of 60.0% and a specificity of 75.9%.

Sub-analysis of patients with hematological cancer revealed that IMV (OR 4.533 [95% CI 2.817–7.294]), sepsis-associated AKI (OR 2.883 [95% CI 1.903–4.368]), HSCT (OR 2.118 [95% CI 1.289–3.482] for autologous, and OR 2.125 [95% CI 1.268–3.559] for allogeneic), and drug-induced AKI (OR 1.727 [95% CI 1.179–2.529]) were predictive factors for in-hospital mortality. A trend towards an association between the need for RRT and in-hospital death was observed (OR 1.451 [95% CI 0.975–2.159]). A corresponding score-point model was created, as shown in Table 8. Upon comparing both models, which demonstrated adequate reliability (*p* = 0.162 and *p* = 0.252, respectively), the AUC values were found to be similar, indicating good discriminability (Figure 2). Using Youden’s index, the optimum cut-off value of 10 was determined as the threshold for predicting in-hospital mortality using this risk score model, with a sensitivity of 71.3% and a specificity of 80.8%.

Figure 3 depicts the cumulative incidence of in-hospital mortality, stratified by the risk score, for the overall population and the subgroup of patients with hematological cancer.

## 4. Discussion

Our analysis revealed that AKI affected 1.0% of cancer patients hospitalized during the study period. This incidence was considerably lower than that reported in previous studies. However, it is important to note that these studies also included cases of mild AKI. In a cross-sectional study by Salahudeen et al., including 3558 patients hospitalized at a Comprehensive Cancer Center, the incidence of AKI was 12%. Based on the RIFLE classification, the corresponding risk, injury, and failure stages were 68%, 21%, and 11%, respectively. With the KDIGO and RIFLE stages showing similar discrimination for hospitalized patients, we may conclude that approximately 4% had severe AKI (corresponding to stages 2 and 3 of KDIGO) [15,16]. Our study reflects the incidence of severe AKI, but it is limited to cases followed by a nephrologist, which may partially explain the substantially lower overall incidence. Another possible reason is that, since 1995, there have been well-defined protocols and algorithms in our institution, which are regularly reviewed according to the state-of-the-art for prophylaxis of AKI secondary to classical chemotherapy (namely platinum-derived agents and ifosfamide), TLS, contrast media, bisphosphonates, and, more recently, ICI and TKI. According to our protocols, most cases of less severe AKI are treated by oncologists. Moreover, another subgroup of patients with AKI stages 1 and 2 could be managed in our nephrological outpatient clinic.

In our study, obstructive AKI emerged as the most prevalent etiology, but a significant decline in its incidence was observed when comparing data from two distinct periods. A case series by Wong et al., involving 102 patients who underwent decompression for malignant ureteral obstruction, reported a success rate of 95% after percutaneous nephrostomy or ureteral stent [17]. This reduction may have been associated with improvements in radiotherapy techniques. Whether obstructive AKI should be managed by nephrologists remains controversial. However, their role is crucial in deciding the need for dialysis prior to nephrostomy, evaluating which kidney requires this procedure, and determining the benefit of bilateral urinary diversion to enable chemotherapy. We favor percutaneous nephrostomy for rapid renal recovery. RRT was performed in only 6% of our patients, owing to an adequate nephrological assessment and a prompt interventional radiology response. The residual cases in which RRT was required before urinary diversion were due to severe hyperkalemia, the need for anticoagulation reversal, or logistic reasons. These patients presented low in-hospital mortality but a poor overall prognosis, with a median survival of six months, as shown in previous results from our group [18].

Increasing age and comorbidities may explain the increased incidence of the other etiologies.

Cancer patients are particularly susceptible to prerenal AKI, as indicated by multiple studies. In our study, patients with gastrointestinal cancer accounted for 35% of the prerenal AKI cases. These patients are exposed to a myriad of hemodynamic insults resulting from nausea, vomiting, diarrhea, and anorexia, predisposing them to volume depletion [19,20].

Nearly 80% of patients treated with anticancer drugs are exposed to potentially nephrotoxic drugs [21]. Our results indicate that less than one-fifth of the cases can be attributed to cancer treatment. Apart from platinum-derived agents, which were primarily responsible for anticancer drug nephrotoxicity in period A, a significant number of cases have been associated with ICI, whose prescription has increased exponentially in recent years (period B) [22]. 

Recent data have reported that the majority of AKI cases in patients receiving ICI are mild to moderate [23]. In conjunction with the fact that there are several recommendations and guidelines for managing immune-related adverse effects, many cases were not evaluated by a nephrologist, suggesting that the incidence of immune-mediated AKI is probably higher. Similar to the general population, antibiotics were the most implicated drug class in our cohort [24].

Sepsis-associated AKI has been established to have the poorest prognosis, with more patients requiring IMV and RRT. This finding is consistent with the existing literature [25,26]. For example, a study conducted by Yang et al. included 356 cancer patients with sepsis who were admitted to the ICU. In the sepsis-associated AKI group, IMV and RRT were significantly higher than those in the non-septic AKI group and were also associated with increased 28-day mortality [27]. CRRT is the most commonly used modality of RRT in the ICU and, as previously shown, it is associated with a much higher mortality risk than IHD.

We identified factors associated with the need for RRT in cancer patients. In univariate analysis, younger age predicted the requirement for RRT. This was also verified in a sub-analysis of patients with hematological cancer, suggesting a likely connection to the higher prevalence of hematological malignancies in this age group. Additional factors contributing to RRT include IMV and hematological malignancies. Critically ill cancer patients are exceptionally susceptible to AKI, and the incidence of requiring RRT varies from 8 to 13% in patients with solid tumors, and 10–34% in patients with hematological malignancies [28]. In clinical practice, decisions regarding the initiation or forgoing of RRT often need to be made before admission to the ICU. After removing IMV from the multivariate model, sepsis stood out as a predictor of the need for RRT. 

The incidence of in-hospital mortality among cancer patients with severe AKI was 26.4% (n = 846). The presence of sepsis, IMV, and RRT, without evidence of an obstructive component, conferred a cumulative incidence of mortality exceeding 70%. In patients with hematological malignancies, HSCT was identified as a predictor of in-hospital mortality. Similar findings were obtained by Parikh et al., who demonstrated that AKI is an independent predictor of overall mortality after non-myeloablative HSCT [29]. The combination of sepsis-associated AKI, drug-induced AKI, and IMV in a patient who underwent HSCT resulted in a cumulative incidence of in-hospital mortality of almost 80%, raising questions about the indication for aggressive life support therapies, such as RRT, in this scenario.

Although our study included a large sample size, several limitations should be considered. First, the retrospective study design has inherent limitations. However, the data were collected prospectively, enabling one of the authors (JMC) to validate the incoming data throughout the years. Second, similar to other retrospective studies, we did not use urine output to detect AKI, as it was not available for all patients. Such patients would have otherwise been categorized as having more severe AKI, which could have underestimated the true incidence. Third, the exclusion of patients without an indication for cardiopulmonary resuscitation and those who were on palliative care may have led to selection biases and immortal time biases in relation to survival analysis. Fourth, we did not account for heterogeneity in cancer stage and disease aggressiveness, important confounders when analyzing predictors of RRT and mortality. This is also true for baseline CKD, a known risk factor for AKI [30]. We may extrapolate from a cohort study conducted in our center by Fernandes et al., published as an abstract in the Portuguese Kidney Journal. The study included 636 patients hospitalized between 2021 and 2022, and it was found that 148 (36.4%) from a total of 407 patients admitted with AKI had previous CKD. Fifth, we did not analyze long-term mortality and CKD development due to the lack of follow-up data. Finally, the scores were developed and tested in the same groups, which might have overestimated overall performance. A study conducted at a specialized tertiary cancer hospital, which exclusively includes severe AKI in patients with nephrological in-hospital follow-up, may hamper the generalizability of the findings, requiring external validation.

## 5. Conclusions

Despite the aforementioned limitations, to the best of our knowledge, this study is one of the few population-based studies to address the epidemiological and clinical features of cancer patients with severe AKI. It introduces an easily calculated risk score that combines acute risk factors to predict in-hospital mortality. This complementary prognostic tool could help in clinical practice with the complex decision to initiate or forgo RRT by identifying high-risk patients.

This study highlights the paradigm shift observed over the last few decades, wherein severe AKI increasingly affects older patients and is less frequently associated with hematological diseases. Notably, a greater number of patients required RRT, but, despite this, overall mortality decreased.

## Figures and Tables

**Figure 1 cancers-16-00561-f001:**
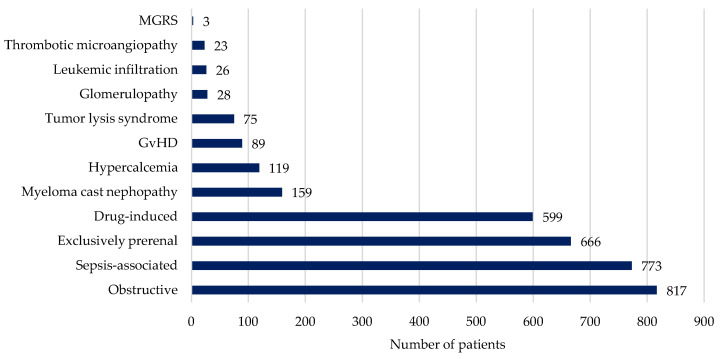
Main etiologies of severe AKI. GvHD: Graft-versus-host disease; MGRS: monoclonal gammopathy of renal significance.

**Figure 2 cancers-16-00561-f002:**
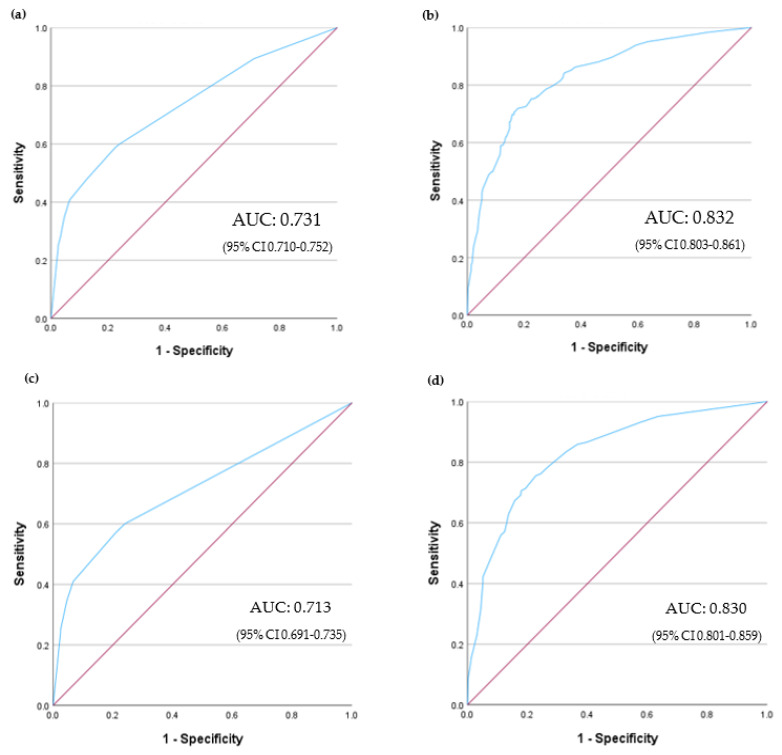
Receiver Operating Characteristic curves and Areas Under the Curve to predict in-hospital death: (**a**) Final model for overall population; (**b**) Final model for the subgroup of patients with hematological cancer; (**c**) Risk model score model for overall population; (**d**) Risk score model for the subgroup of patients with hematological cancer. AUC: Area under the curve; CI: Confidence Interval.

**Figure 3 cancers-16-00561-f003:**
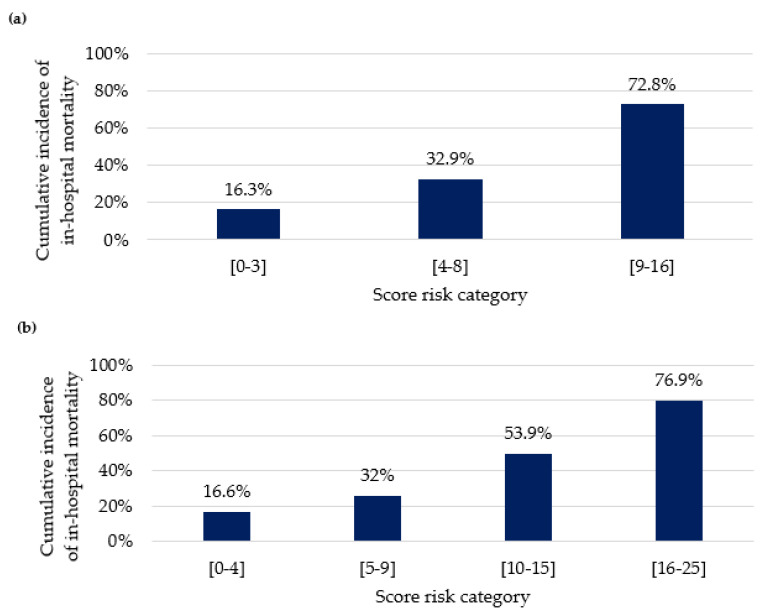
(**a**) Cumulative incidence of in-hospital mortality stratified by risk score for the overall population; (**b**) Cumulative incidence of in-hospital mortality stratified by risk score for the subgroup of patients with hematological cancer.

**Table 1 cancers-16-00561-t001:** Patients’ characteristics from Period A and Period B.

Characteristics	Overall	Period A (1995–2010)	Period B (2011–2023)	*p*-Value
Patients	3201	1588	1613	
Male	1836 (57.4)	852 (53.7)	984 (61.0)	<0.001
Age, mean ± SD	62.5 ± 17.2	59.2 ± 19.2	65.7 ± 14.6	<0.001
≥40 years	2880 (90.0)	1360 (85.5)	1523 (94.2)	<0.001
<40 years	321 (10.0)	228 (14.4)	93 (5.8)	
Multiple primary tumors	96 (3.0)	32 (2.0)	64 (4.0)	0.001
Cancer				
Hematological	779 (24.3)	466 (29.3)	313 (19.4)	<0.001
HSCT	284 (36.4)	172 (36.9)	112 (35.8)	<0.001
Allo	133 (17.1)	77 (4.8)	54 (3.3)	
Auto	148 (19.0)	95 (6.0)	58 (3.6)	
Solid	2422 (75.7)	1122 (70.7)	1300 (80.6)	<0.001
Gastrointestinal	749 (30.9)	319 (28.4)	430 (33.1)	
Gynecological	571 (23.5)	224 (20.0)	347 (26.7)	
Urological	438 (18.1)	287 (25.6)	151 (11.6)	
Head and neck	172 (7.1)	87 (7.8)	85 (6.5)	
Breast	128 (5.3)	79 (7.0)	49 (3.8)	
Lung	123 (5.1)	31 (2.8)	92 (7.1)	
Sarcoma	115 (4.7)	49 (4.4)	66 (5.1)	
AKI etiology				
Exclusively prerenal	666 (20.8)	277 (17.6)	389 (24.1)	<0.001
Sepsis-associated	773 (24.1)	308 (19.4)	465 (28.8)	<0.001
Drug-induced	599 (18.7)	265 (16.8)	334 (20.7)	0.006
Immunotherapy	52 (1.6)	0	52 (3.2)	<0.001
Obstructive	889 (27.8)	503 (32.0)	314 (19.5)	<0.001
Pos-operative AKI	321 (10)	187 (11.8)	134 (8.3)	0.001
Outcomes				
ICU admission	637 (19.9)	326 (20.5)	311 (19.3)	0.377
IMV	423 (13.2)	233 (14.7)	190 (11.8)	0.016
RRT	539 (16.8)	279 (17.6)	260 (22.3)	0.002
CRRT	364 (11.4)	165 (10.4)	199 (12.3)	
IHD	310 (9.7)	131 (8.2)	179 (11.1)	
Mortality	846 (26.4)	464 (29.2)	382 (23.7)	<0.001

Values are n (%) unless specified otherwise. AKI: acute kidney injury; CRRT: continuous renal replacement therapy; HSCT: hematopoietic stem cell transplantation; ICU: intensive care unit; IHD: intermittent hemodialysis; IMV: invasive mechanical ventilation; RRT: renal replacement therapy; SD: standard deviation.

**Table 2 cancers-16-00561-t002:** Patients’ characteristics per AKI etiology.

Characteristics		Obstructive n = 817 (24.2%)	Sepsis-Associated n = 773 (22.9%)	Prerenal n = 666 (19.7%)	Drug-Induced n = 599 (17.7%)
Male sex		403 (49.3)	475 (61.4)	385 (57.8)	344, (57.4)
Age, mean ± SD		65.4 ± 14.6	59.5 ± 18.1	68.0 ± 13.2	59.1 ± 19.4
Most common neoplasms		Urological 269 (32.9)	Hematological 336 (43.5)	Gastrointestinal 235 (35.3)	Hematological 242 (40.4)
		Gynecological 260 (31.8)	Gastrointestinal 152 (19.7)	Urological 121 (18.2)	Gastrointestinal 99 (16.5)
		Gastrointestinal 202 (24.7)	Urological 109 (14.1)	Gynecological 72 (10.8)	Head and neck 60 (10.0)
ICU admission		18 (2.2)	454 (58.7)	54 (8.1)	164 (27.4)
Outcomes	IMV	6 (0.7)	365 (47.2)	45 (6.8)	129 (21.5)
	RRT	49 (6.0)	355 (45.9)	69 (10.4)	170 (28.4)
	Mortality	110 (13.5)	406 (52.5)	131 (19.7)	329 (32.9)

Values are n (%) unless specified otherwise. AKI: acute kidney injury; ICU: intensive care unit; IMV: invasive mechanical ventilation; RRT: renal replacement therapy; SD: standard deviation.

**Table 3 cancers-16-00561-t003:** Predictors of RRT among patients hospitalized with severe AKI in the final logistic regression model.

Factors	Multivariate Analysis		
	B Coefficient	OR (95% CI)	*p*-Value
IMV			
No	-	Ref.	
Yes	2.008	7.450 (5.455–10.174)	<0.001
Type of cancer			
Solid	-	Ref.	
Hematological	0.844	2.325 (1.849–2.925)	<0.001
Sepsis-associated AKI			
No	-	Ref.	
Yes	0.246	1.279 (0.980–1.670)	0.071
Prerenal AKI			
No	-	Ref.	
Yes	−3.968	0.019 (0.006–0.060)	<0.001
Obstructive AKI			
No	-	Ref.	
Yes	−1.083	0.339 (0.251–0.457)	<0.001

AKI: acute kidney injury; CI: confidence interval; IMV: invasive mechanical ventilation; OR: Odds ratio.

**Table 4 cancers-16-00561-t004:** Predictors of RRT among patients hospitalized with severe AKI in the final logistic regression model (excluding IMV).

Factors	Multivariate Analysis		
	B Coefficient	OR (95% CI)	*p*-Value
Type of cancer			
Solid	-	Ref.	
Hematological	1.267	3.552 (2.886–4.372)	<0.001
Sepsis-associated AKI			
No	-	Ref.	
Yes	1.111	3.037 (2.462–3.746)	<0.001
Prerenal AKI			
No	-	Ref.	
Yes	−3.939	0.019 (0.006–0.061)	<0.001
Obstructive AKI			
No	-	Ref.	
Yes	−1.110	0.330 (0.246–0.442)	<0.001

AKI: acute kidney injury; CI: confidence interval; IMV: invasive mechanical ventilation; OR: Odds ratio.

**Table 5 cancers-16-00561-t005:** Predictors of RRT among the subgroup of hospitalized patients with hematological cancer and severe AKI in the final logistic regression model.

Factors	Multivariate Analysis		
	B Coefficient	OR (95% CI)	*p*-Value
IMV			
No	-	Ref.	
Yes	2.445	11.533 (7.954–16.723)	<0.001
Age			
≥40 years	-	Ref.	
<40 years	0.406	1.501 (1.001–2.250)	0.049
Prerenal AKI			
No	-	Ref.	
Yes	−2.886	0.056 (0.013–0.247)	<0.001
Obstructive AKI			
No	-	Ref.	
Yes	−0.782	0.457 (0.245–0.854)	0.014

AKI: acute kidney injury; CI: confidence interval; IMV: invasive mechanical ventilation; OR: Odds ratio; RRT: renal replacement therapy.

**Table 6 cancers-16-00561-t006:** Associations between predictor variables and in-hospital mortality among patients hospitalized with severe AKI.

Characteristics	Univariate Analysis				
	Overall n = 3201	Survivors n = 2355	Non-Survivors n = 846	OR (95% CI)	*p*-Value
Sex					
Female	1365 (42.6)	1030 (43.7)	335 (39.6)	Ref.	
Male	1836 (57.4)	1325 (56.3)	511 (60.4)	1.186 (1.010–1.392)	0.037
Age					
≥40 years	2880 (90.0)	2150 (91.3)	730 (86.3)	Ref.	
<40 years	321 (10.0)	205 (8.7)	116 (13.7)	1.667 (1.308–2.124)	<0.001
Multiple primary tumors					
No	3105 (97.0)	2285 (97.0)	820 (96.9)	Ref.	
Yes	96 (3.0)	70 (3.0)	26 (3.1)	1.035 (0.655–1.635)	0.883
Type of cancer					
Solid	2422 (75.7)	1903 (80.8)	519 (61.3)	Ref.	
Hematological	779 (24.3)	452 (19.2)	327 (38.7)	2.653 (2.233–3.151)	<0.001
ICU admission					
No	2564 (80.1)	2113 (89.7)	451 (53.3)	Ref.	
Yes	637 (19.9)	242 (10.3)	395 (46.7)	7.647 (6.327–9.243)	<0.001
IMV					
No	2778 (86.8)	2235 (94.9)	543 (64.2)	Ref.	
Yes	423 (13.2)	120 (5.1)	303 (35.8)	10.393 (8.247–13.097)	<0.001
AKI etiology					
Exclusively prerenal					
No	2535 (79.2)	1820 (77.3)	715 (84.5)	Ref.	
Yes	666 (20.8)	535 (22.7)	131 (15.5)	0.623 (0.505–0.769)	<0.001
Sepsis-associated					
No	2428 (75.9)	1988 (84.4)	440 (52.0)	Ref.	
Yes	773 (24.1)	367 (15.6)	406 (48.0)	4.998 (4.196–5.954)	<0.001
Drug-induced					
No	2602 (81.3)	1953 (82.9)	649 (76.7)	Ref.	
Yes	599 (18.7)	402 (17.1)	197 (23.3)	1.475 (1.217–1.787)	<0.001
Obstructive					
No	2312 (72.2)	1593 (67.6)	719 (85.0)	Ref.	
Yes	889 (27.8)	762 (32.4)	127 (15.0)	0.369 (0.300–0.454)	<0.001
Postoperative AKI					
No	2877 (89.9)	2125 (90.2)	752 (88.9)	Ref.	
Yes	324 (10.1)	230 (9.8)	94 (11.1)	1.144 (0.888–1.475)	0.298
Need for RRT					
No	2562 (80.0)	2066 (87.7)	496 (58.6)	Ref.	
Yes	639 (20.0)	289 (12.3)	350 (41.4)	5.045 (4.197–6.064)	<0.001
Hematological malignancy					
HSCT					
No	498 (63.9)	336 (74.3)	162 (49.5)	Ref.	
Auto	148 (19.0)	65 (14.4)	83 (25.4)	2.648 (1.820–3.853)	<0.001
Allo	133 (17.1)	51 (11.3)	82 (25.1)	3.335 (2.243–4.958)	
GvHD					
No	694 (89.1)	415 (91.8)	279 (85.3)	Ref.	
Yes	85 (10.9)	37 (8.2)	48 (14.7)	1.930 (1.224–3.041)	<0.001

Values are n (%). AKI: acute kidney injury; CI: confidence interval; GvHD: Graft-versus-host-disease; HSCT: hematopoietic stem cell transplantation; ICU: intensive care unit; IMV: invasive mechanical ventilation; OR: Odds ratio; RRT: renal replacement therapy.

**Table 7 cancers-16-00561-t007:** Predictors of in-hospital mortality among patients hospitalized with severe AKI in the final logistic regression model with score points.

Factors	Multivariate Analysis			
	B Coefficient	OR (95% CI)	*p*-Value	Score Points
IMV				
No	-	Ref.		
Yes	1.336	3.804 (2.844–5.089)	<0.001	8
Sepsis-associated AKI				
No	-	Ref.		
Yes	0.772	2.164 (1.731–2.706)	<0.001	4
Need for RRT				
No	-	Ref.		
Yes	0.716	2.046 (1.640–2.552)	<0.001	4
Obstructive AKI				
No	-	Ref.		
Yes	−1.537	0.647 (0.518–0.809)	<0.001	−1

AKI: acute kidney injury; CI: confidence interval; IMV: invasive mechanical ventilation; OR: Odds ratio; RRT: renal replacement therapy.

**Table 8 cancers-16-00561-t008:** Predictors of in-hospital mortality among the subgroup of hospitalized patients with hematological cancer and severe AKI in the final logistic regression model with score points.

Factors	Multivariate Analysis			
	B Coefficient	OR (95% CI)	*p*-Value	Score Points
IMV				
No	-	Ref.		
Yes	1.511	4.533 (2.817–7.294)	<0.001	9
Sepsis-associated AKI				
No	-	Ref.		
Yes	1.059	2.883 (1.903–4.368)	<0.001	6
HSCT				
No	-	Ref.		
Auto	0.751	2.118 (1.289–3.482)	0.003	4
Allo	0.754	2.125 (1.268–3.559)	0.004	4
Need for RRT				
No	-	Ref.		
Yes	0.372	1.451 (0.975–2.159)	0.067	3
Drug-induced AKI				
No	-	Ref.		
Yes	0.546	1.727 (1.179–2.529)	0.005	3

AKI: acute kidney injury; CI: confidence interval; HSCT: hematopoietic stem cell transplantation; IMV: invasive mechanical ventilation; OR: Odds ratio; RRT: renal replacement therapy.

## Data Availability

The datasets generated and/or analyzed during the current study are not publicly available. However, they are available from the corresponding author upon reasonable request.

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
