# Peer review of "Severe Acute Kidney Injury in Hospitalized Cancer Patients: Epidemiology and Predictive Model of Renal Replacement Therapy and In-Hospital Mortality"

_cancers, 2024, doi:10.3390/cancers16030561_

Round 1
Reviewer 1 Report
Comments and Suggestions for Authors
The paper is interesting, and brings a very large case history.
I have few comments.
An aspect that deserves to be discussed more thoroughly is linked to the typology of patients, which are limited to those who have come to the attention of the nephrologist. I wish the authors would discuss this more thoroughly. Have they seen any differences in outcomes (type of mortality, need for RRT) compared to case series of patients reported by oncologists?
good
Reviewer 2 Report
Comments and Suggestions for Authors
This study retrospectively analyzed data on 3,201 cancer patients with severe AKI who were hospitalized at a Comprehensive Cancer Center between 1995 and 2023 and received inpatient nephrology follow-up. The objectives were to determine the incidence and causes of severe AKI, identify risk factors associated with the need for RRT and in-hospital mortality, and develop predictive models.
Key findings include: 1) 1% of hospitalized cancer patients developed severe AKI, most commonly from obstructive (24.2%), sepsis-associated (22.9%), prerenal (19.7%), and drug-induced (17.7%) causes; 2) 20% of patients required RRT and 26.4% died during hospitalization, with sepsis, IMV, and RRT increasing mortality risk; 3) Predictive models were developed to estimate the likelihood of requiring RRT and dying in the hospital, with good discrimination ability; 4) Risk score models were also derived to quantify mortality risk based on a simple points system. The study provides valuable real-world data and prognostic tools to guide clinical decision-making for this complex patient population.
Comments:
As a retrospective study relying on existing documentation, this study design has inherent limitations compared to a prospective controlled study. There was likely less control and oversight on data quality, with greater potential for missing data. Variability in measurement and testing practices over 28 years may have impacted classifications. Researchers had to rely fully on the accuracy and completeness of clinical documentation for case identification, etiological assessments, and outcome determinations. These issues introduce the possibility of information biases. Additionally, the exclusion of patients not deemed appropriate for full code or who were on palliative care only opens the door for selection biases and immortal time bias with regards to survival analyses.
Being a single-center study from a specialized tertiary cancer hospital may hamper the generalizability and external validity of the findings. The incidence rates, predictor analysis results, and prognostic model performance may not directly translate to other institutions or populations without external validation. As the sole cancer center for a region, referral patterns and case mixes may differ from community settings.
By not incorporating urine output criteria along with creatinine for identifying and staging AKI, some cases likely suffered from misclassification given urine output sometimes detects injury earlier. Such patients would have otherwise been categorized as having more severe AKI. This could underestimate the true incidence rates. On the other end of the timeline, lacking longer-term outcome data on renal function recovery, post-discharge mortality, and CKD development limits understanding of the lasting impacts of AKI in cancer patients. Survivorship beyond hospitalization remains unclear.
Not accounting for heterogeneity in cancer stage and disease aggressiveness is an important confounder when analyzing predictors of RRT and mortality. More extensive, metastatic, or rapidly progressive cancers independently worsen prognosis regardless of AKI. Thus some predictors may have been influenced by this factor not being incorporated. Similarly, baseline chronic kidney disease is a pertinent risk factor for developing AKI and portending worse outcomes. The possible confounding of this variable also being absent from predictive modeling should be considered.
Reviewer 3 Report
Comments and Suggestions for Authors
Authors decribed the epidermiology and predictive model for RRT and mortality with severe AKI in cancer patients.
Very interesting, however I have some questions and recommend correction.
1. Abstract : Line 34 . RTT -> RRT
2. Why did you analyzed in two timelime?
Just only 1 table described the differences.
All results were outcomes associated with AKI
If any reason for two timeline, describe the reason in the method section. And re-analize according to two timeline.
